# Radiation tolerance of two-dimensional material-based devices for space applications

Tobias Vogl [1], Kabilan Sripathy [1], Ankur Sharma [2], Prithvi Reddy [3], James Sullivan [4], Joshua R. Machacek [4], Linglong Zhang[2], Fouad Karouta[5], Ben C. Buchler [1], Marcus W. Doherty [3], Yuerui Lu[2] & Ping Koy Lam [1]

Characteristic for devices based on two-dimensional materials are their low size, weight and power requirements. This makes them advantageous for use in space instrumentation, including photovoltaics, batteries, electronics, sensors and light sources for long-distance quantum communication. Here we present a comprehensive study on combined radiation effects in Earth's atmosphere on various devices based on these nanomaterials. Using theoretical modeling packages, we estimate relevant radiation levels and then expose field-effect transistors, single-photon sources and monolayers as building blocks for future electronics to $\gamma$-rays, protons and electrons. The devices show negligible change in performance after the irradiation, suggesting robust suitability for space use. Under excessive $\gamma$-radiation, however, monolayer $WS_2$ shows decreased defect densities, identified by an increase in photoluminescence, carrier lifetime and a change in doping ratio proportional to the photon flux. The underlying mechanism is traced back to radiation-induced defect healing, wherein dissociated oxygen passivates sulfur vacancies.

[1] Centre for Quantum Computation and Communication Technology, Department of Quantum Science, Research School of Physics and Engineering, The Australian National University, Acton, ACT 2601, Australia. [2] Research School of Engineering, The Australian National University, Acton, ACT 2601, Australia. [3] Laser Physics Centre, Research School of Physics and Engineering, The Australian National University, Acton, ACT 2601, Australia. [4] Plasma Research Laboratory, Research School of Physics and Engineering, The Australian National University, Canberra, ACT 2601, Australia. [5] Australian National Fabrication Facility, Research School of Physics and Engineering, The Australian National University, Acton, ACT 2601, Australia. Correspondence and requests for materials should be addressed to T.V. (email: tobias.vogl@anu.edu.au) or to P.K.L. (email: ping.lam@anu.edu.au)

In the near future, quantum tunneling will set a hard limit to further miniaturization of silicon-based electronics. Research on alternative materials, however, demonstrated fabrication beyond this limit[1,2]. Of particular interest are monolayered two-dimensional (2D) materials such as graphene[3] and transition metal dichalcogenides (TMDs) of the form of $MX_2$ (M = Mo, W and X = S, Se)[4]. Record electron mobility in 2D materials has enabled multiple technology demonstrations of atomically thin field-effect transistors (FETs)[5–9]. Furthermore, due to their semiconducting bandstructure, TMDs have applications in optoelectronics and photonics[10]. Their intrinsically low size, weight, and power requirements and chemical stability make 2D material-based devices a promising candidate for space instrumentation. Beyond integrated electronics, 2D materials in space technology can be utilized for solar cells[11], batteries[12], sensors as well as non-classical light sources for long-distance quantum communication[13]. The quantum emission from point defects in 2D materials have desirable properties for single-photon sources, as they can be easily integrated with photonic networks, have an intrinsic out-coupling efficiency of unity, and offer long-term stable, high-luminosity single-photons at room temperature (RT)[13–17]. An ideal single-photon source can enhance the data communication rates of satellite-based quantum key distribution[18].

While 2D materials offer great opportunities for space missions, their current low technological readiness level (TRL) restricts deployment (current state-of-the-art is TRL 3–4). In addition to further device development, 2D materials need to be certified for the harsh conditions of space. Space qualification studies usually consist of vacuum and thermal cycling, vibration, and shock tests as well as exposure to radiation[19]. Vibration or shock will not pose a threat for nanomaterials and vacuum and thermal cycling is routinely done in experiments[20–22]. Of particular interest, however, is the effect of radiation on 2D materials. While radiation effects on the electrical properties of graphene have been studied extensively[23–28], less is known about these effects on TMDs and other 2D materials[29–33]. In particular, no study investigates the effect of radiation on optical characteristics of 2D materials. Moreover, there exists no comprehensive study on the effects of combined radiation types on properties of various devices in the context of space certification. The damage caused by high-energy particles and γ-rays is of major concern for all spacecraft, especially as weight restrictions limit shielding options. While testing directly in a space environment as planned for graphene is possible[34], a more practical way is to replicate space environments on Earth.

As already mentioned, single-photon sources based e.g. on defects in hexagonal boron nitride (hBN) and FETs based on monolayers are useful for space applications. These devices are influenced by their electrical and/or optical properties (quantum emitters in hBN are dependent on the piezoelectric environment of hBN[35]). It is possible that low-energy radiation on the order of the bandgap (~1–10 eV) could change the charge state of defects, causing them to enter a dark state, usually for a finite time. High-energy radiation, on the other hand, could create new defects in the crystal lattice. If close to the quantum emitter, the new defects could either change the charge state permanently or create a second independent emitter nearby. In any of these scenarios, the single-photon source would become unusable. With respect to the FETs, the radiation could change the carrier density, which alters their performance.

Here we present a comprehensive study on the effects of radiation in the atmosphere on various devices based on 2D materials. We start with modeling radiation levels in the thermosphere using the SPace ENVironment Information System (SPENVIS), software provided by the European Space Agency[36]. With the knowledge from the simulations, we expose our devices

to the most common radiation types in orbit: gamma-rays as well as energetic protons and electrons. We look at isolated effects and combined effects by exposing devices to all three types of radiation. For each test, we fully characterize all devices back to back, shortly before and after the exposure. At radiation levels common for satellite altitudes up to geostationary orbit, no changes in the characteristics of the 2D materials are observed. However, under excessive γ-irradiation, $WS_2$ monolayers exhibit significant change in its optical emission. By studying the effects of oxygen plasmas and γ-irradiation in different atmospheres, the mechanism is traced back to oxygen-related vacancy healing. Additional density functional theory (DFT) calculations show that charge trapping states disappear after the healing, thus explaining the change in optical emission.

## Results

**Radiation levels in orbit.** The Earth is protected from solar wind and cosmic particles by its magnetic field. As a result, high-energy protons and electrons are trapped on trajectories oscillating between both magnetic poles in the so-called Van Allen belts. While essential for life on Earth, the trapped particle belts pose great threat to any spacecraft orbiting through these particle belts. Near the magnetic poles, the inner belt can extend down to altitudes of 200 km. Owing to misalignment of the magnetic dipole and rotation axis of the Earth, this appears as the South Atlantic Anomaly (SAA; see Fig. 1a, b). Because of this inhomogeneity, the total radiation dosage is strongly dependent on the orbital inclination. Thus we calculated the particle spectra for different spacecraft trajectories with inclinations of 20° (here defined as equatorial orbit), 51.6° (orbit of the International Space Station (ISS)), and 98° (here defined as polar orbit) for 500 km altitude and average over the full orbit. In general, the energy spectrum for protons in low Earth orbit (LEO) ranges from 100 keV to 400 MeV, while for electrons it ranges from 40 keV to 7 MeV. Low-energy particles are typically absorbed by the walls of the spacecraft, which acts as a non-ideal high-pass filter. High-energy ions, however, loose energy during their interaction with the shielding material and thus the lower ends of the spectra are always finite unless every charge carrier is stopped (e.g. for thick shielding). The shielded flux spectra for protons and electrons after 1.85 mm of Al shielding and integrated over a 1-year mission is shown in Fig. 1c, d. Surprisingly, the polar orbit does not have the highest fluence, as spacecrafts with 51.6° inclination spend more time in the SAA than spacecrafts with 98° inclination, similarly for protons and electrons. A spectral distribution with an absence of low-energy protons, as shown in Fig. 1c, is advantageous, because only low-energy particles can deposit significant amounts of energy into the payload. It should be mentioned that the electrons in Fig. 1d do not originate from trapped electrons in the Van Allen belt but rather are secondary electrons produced via ionizing interactions of high-energy protons with the Al atoms in the shielding material.

While the particle fluence spectra are directly accessible through SPENVIS, similar tools for γ-rays do not exist. Gamma-rays mostly originate either directly from the sun or from the radioactive decay of trapped particles. For our study, we use data from the CORONAS-I satellite[37,38], which mapped the γ-ray flux above the Earth at 500 km altitude (see Table 1).

**Device fabrication and characterization.** Exfoliated hBN flakes have been treated with an oxygen plasma and successively rapidly thermally annealed[14]. The oxygen plasma creates point defects in the crystal lattice, which act as trapping sites for localized excitons. The single-photon emitters formed in this way were located and characterized using a confocal microphotoluminescence

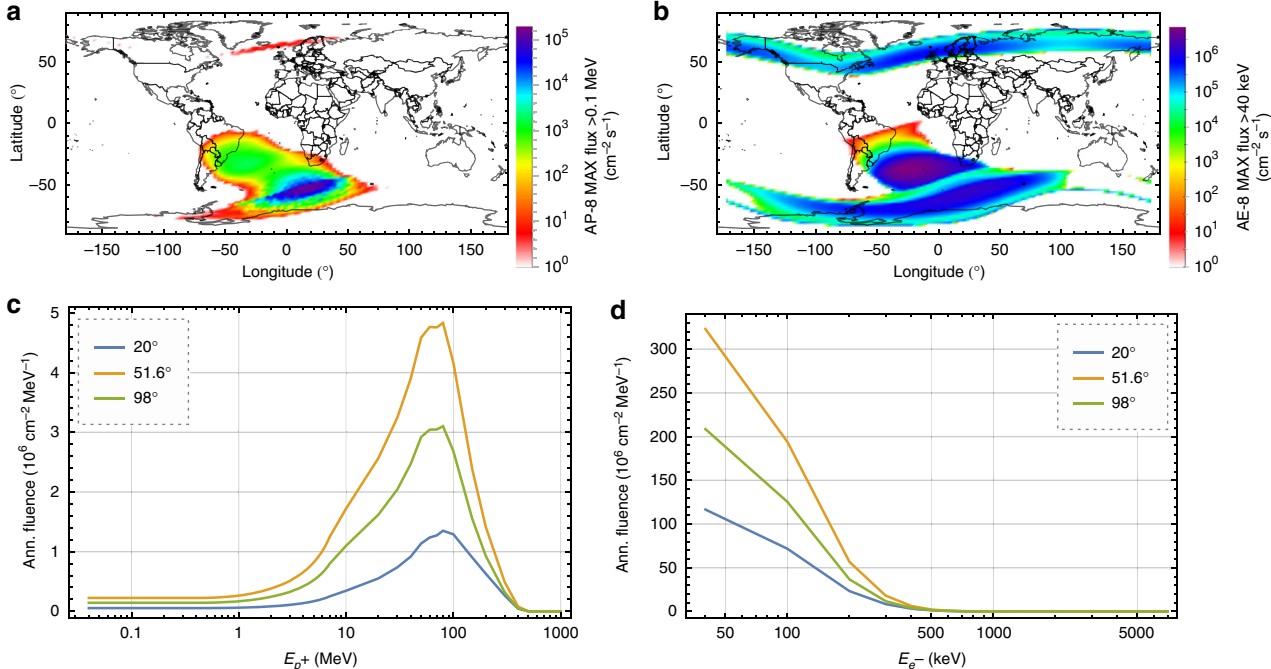

**Fig. 1** Space environment. Geographical distribution of the trapped **a** proton and **b** electron flux at 500 km altitude, calculated with the AP-8 MAX and AE-8 MAX models in SPENVIS, respectively. Integrated annual **c** proton and **d** electron fluence after 1.85 mm of Al shielding for typical orbital inclinations

| Table 1 Summary of total measured gamma-ray flux from the CORONAS-I satellite at 500 km altitude | | |
|---|---|---|
| Location | γ-Ray energy [MeV] | Flux [$cm^{-2} s^{-1} sr^{-1} MeV^{-1}$] |
| Equator | 0.32–1 | 0.079(59) |
| Equator | 1–3 | 0.022(14) |
| Polar cap | 0.32–1 | 0.174(59) |
| Polar cap | 1–3 | 0.095(14) |
| Full data available in ref. [38] | | |

(µPL) system equipped with an ultrashort-pulsed laser for time-resolved measurements (see Methods). The emitters are excited off-resonantly at 522 nm, less than half of the bandgap of hBN ($E_g = 6$ eV[39]), preventing two-photon absorption. For the quantum emitters, we measured the spectrum, excited state lifetime, and second-order correlation function (see Methods).

The atomically thin FET devices (see Fig. 2a) were characterized by their device ON/OFF performance using the standard back gate sweep from −60 to +60 V at different biases between source and drain. Also, the conventional performance I–V curves of the device were recorded at various back gate voltages in the ON regime of the functional FET device.

Since 2D materials have often been proposed as candidates for the post-silicon age, we also tested monolayer TMDs in their native state as basic building blocks for future electronics and optoelectronics. After transfer to a Si/SiO₂ substrate (see Fig. 2b), the monolayer thickness is confirmed by phase-shift interferometry (PSI), with the corresponding PSI image shown in Fig. 2c. In this case, the WS₂ crystal has an optical path length difference (OPD) of 17.7 nm. With rigorous coupled-wave analysis simulations[40], the OPD can be converted to a physical thickness of 0.66 nm, matching well atomic force microscopic measurements[41]. We characterized each flake optically with the µPL set-up in terms of emission spectrum (averaged over the full monolayer), carrier lifetime, and power saturation. The carrier lifetime data, deconvoluted from the system response, is fitted with a

bi-exponential from which radiative and non-radiative decay time $\tau_r$ and $\tau_{nr}$ can be extracted. Every flake is scanned with a 1-µm grid and a spectrum is recorded at each point. To gather enough statistics, a total of 49 monolayer flakes with areas ranging from 60 to 1290 µm² have been characterized. Unless stated otherwise, all optical and electrical measurements have been carried out at RT. More than 80 devices were investigated throughout this study; herein we only show exemplary results and average over the full data set (see Supplementary Notes 2, 5, and 6).

**Gamma-ray tests**. The γ-ray source predominantly used for space qualification is the radioactive isotope $^{60}_{27}$Co, which emits photons with energies of 1.17 and 1.33 MeV as it decays. Owing to availability, we used the isotope $^{22}_{11}$Na instead, which decays into $^{22}_{10}$Ne via the emission of a 1.28 MeV photon[42], similar to the γ-ray energy from the Co isotope. From its initial nominal activity of 1.04 GBq, a total photon flux of 10.3 MBq $cm^{-2} sr^{-1} MeV^{-1}$ is emitted into the output mode of the Tungsten container in which the source was kept. We placed the samples at distances of $d = 10.0(1)$, 13.0(1), 18.0(1), and 40.0(1) cm from the source output, thus simulating various altitudes and times in orbit (see Supplementary Fig. 1). All samples were irradiated for 2:27 h, meaning that the maximal fluence at the closest distance to the source was $F_\gamma = 18.41 \times 10^9$ $cm^{-2} sr^{-1} MeV^{-1}$. Unless stated otherwise, the crystals presented in this section were irradiated with the highest photon flux. Unexposed control samples ensured that any potential changes are solely due to irradiation.

The performance of the single-photon emitters in hBN and the FET devices remained invariant when comparing samples before and after the γ-ray exposure. The zero phonon line (ZPL) of a sample quantum emitter as shown in Fig. 3a peaked initially at 563.78(8) nm with a linewidth of 4.29(13) nm (extracted from fit). Unless stated otherwise, all uncertainties are 95% confidence intervals. After the crystal was irradiated, the ZPL peaked at 563.79(13) nm with a linewidth of 4.73(19) nm. Similarly, its $g^{(2)}(0)$ did not change (see Fig. 3b) with $g_i^{(2)}(0) = 0.185(23)$ and $g_f^{(2)}(0) = 0.188(25)$, where index i and f stand for before and

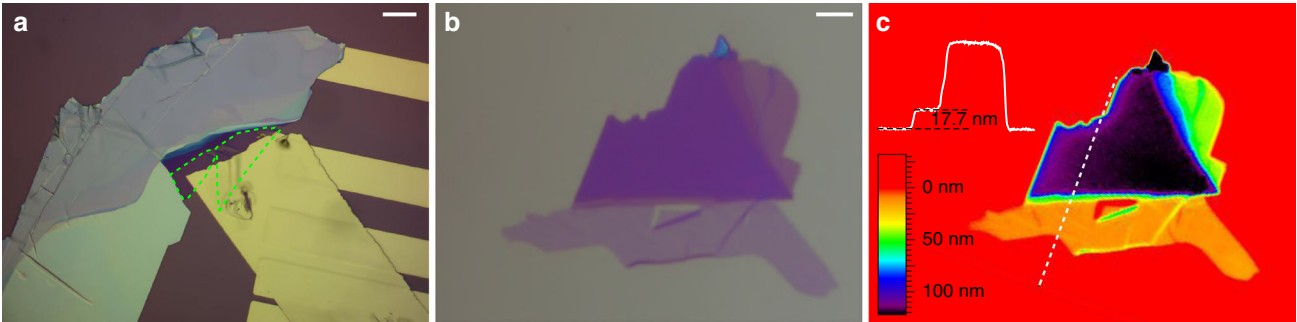

**Fig. 2** Fabrication. **a** Microscopic image of a $MoS_2$ field-effect transistor device under ×500 magnification. The monolayer is framed within the green dashed line. **b**, **c** Microscopic and phase-shift interferometric images of the $WS_2$ monolayer presented in the main text. The inset in **c** shows the optical path length difference (OPD) along the white dashed line. The monolayer has an OPD of 17.7 nm, which corresponds to a physical thickness of 0.66 nm. The scale bars in **a**, **b** are 20 and 5 μm, respectively

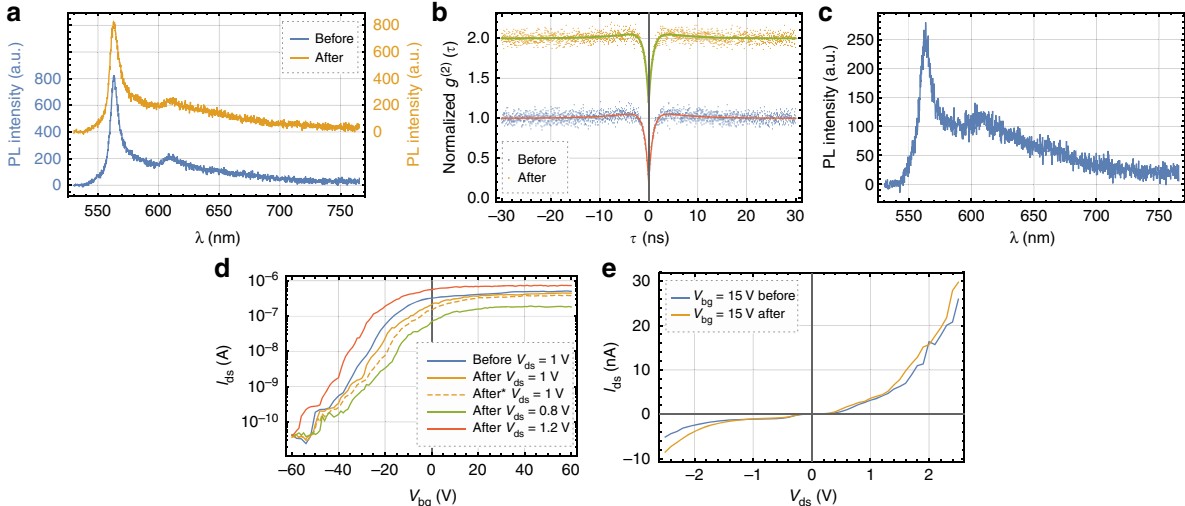

**Fig. 3** γ-Ray tests of two-dimensional material-based devices. **a** Photoluminescence spectra of an hexagonal boron nitride quantum emitter before and after the γ-ray tests show no changes (vertically offset for clarity). **b** Second-order correlation function dipping at zero time delay to 0.185(23) before and to 0.188(25) after the irradiation. The values were obtained from fitting a three-level system (for clarity, the fits are differently colored). **c** Spectrum of a newly created quantum emitter after the γ-ray test. As the emitter was not annealed following the irradiation, its brightness and stability was lacking behind plasma-etched and annealed emitters. **d** Back gate sweeps before and after the irradiation with different drain-source biases $V_{ds}$. The orange dashed line was recorded 5 h past the orange solid line to check for temporal variations. In terms of current ON/OFF ratio, the temporal variations are larger than the variations caused by the γ-rays. Tuning the $V_{ds}$ can restore the initial performance (see area between green and red lines). **e** The $I–V$ curve at a fixed $V_{bg} = $ 15 V shows only slight alteration after irradiation. The shift of the threshold voltage is <0.1 V

after the exposure, respectively. While the quantum emitters already present in the hBN crystal did not change with respect to their optical emission properties, the γ-rays were able to create five new emitters on ≈40,000 μm$^2$ of crystal area. Thus the probability of creating a second emitter directly adjacent to another is very low. Figure 3c shows the spectrum of one of the newly created emitters. As the crystal was not subsequently annealed, its brightness as well as stability was not as good as for other emitters[14].

Comparably, the FETs were also only marginally affected by the γ-rays. Figure 3d shows back gate sweeps for a $MoS_2$ transistor. The current ON/OFF ratio $\beta$ was reduced from $\beta_i = $ 21,213 to $\beta_f = $ 14,863 at a drain-source bias of $V_{ds} = $ 1 V. While this is a significant change in the ON/OFF ratio, we measured the ON/OFF ratio 5 h later and saw $\beta$ further reduced to 11,781 (dashed line in Fig. 3d). In fact, the standard deviation of the variations on control samples as well as variations before and after irradiation were roughly 4000. Hence, we attribute these changes in the ON/OFF ratio to temporal variations only. The FETs in general are sensitive to surface adsorption, which causes

these temporal variations. In addition, $I–V$ characteristics are highly dependent on the Schottky or contact resistance, which varies across different measurements. However, by varying the drain-source bias from 0.8 to 1.2 V, the initial performance could be restored (see Fig. 3d). Another characteristic of transistors is the $I–V$ curve measured at fixed back gate voltages $V_{bg}$. For $V_{bg} = $ 15 V, this is shown in Fig. 3e and for other $V_{bg}$ in Supplementary Fig. 2. The $I–V$ curves show no change due to the irradiation. The FETs, while conducting at a particular back gate bias, show no threshold voltage shift with $\Delta V_{th} < 0.1$ V.

While the 2D material-based devices showed no change after the γ-ray tests, the optical signature of monolayer $WS_2$ changed remarkably: The monolayer shown in Fig. 2b showed a significant increase in photoluminescence (PL). Moreover, the brightness increased by a factor of 2.99 after being exposed to the γ-rays (see Fig. 4a, b). Furthermore, from the averaged PL spectrum of the monolayer (see Fig. 4c, for details of the averaging algorithm see Supplementary Note 1), we extract that the exciton/trion ratio $\alpha$ changed from 0.706(11) to 1.138(19). Both the exciton and trion emission were enhanced, however, the exciton emission was

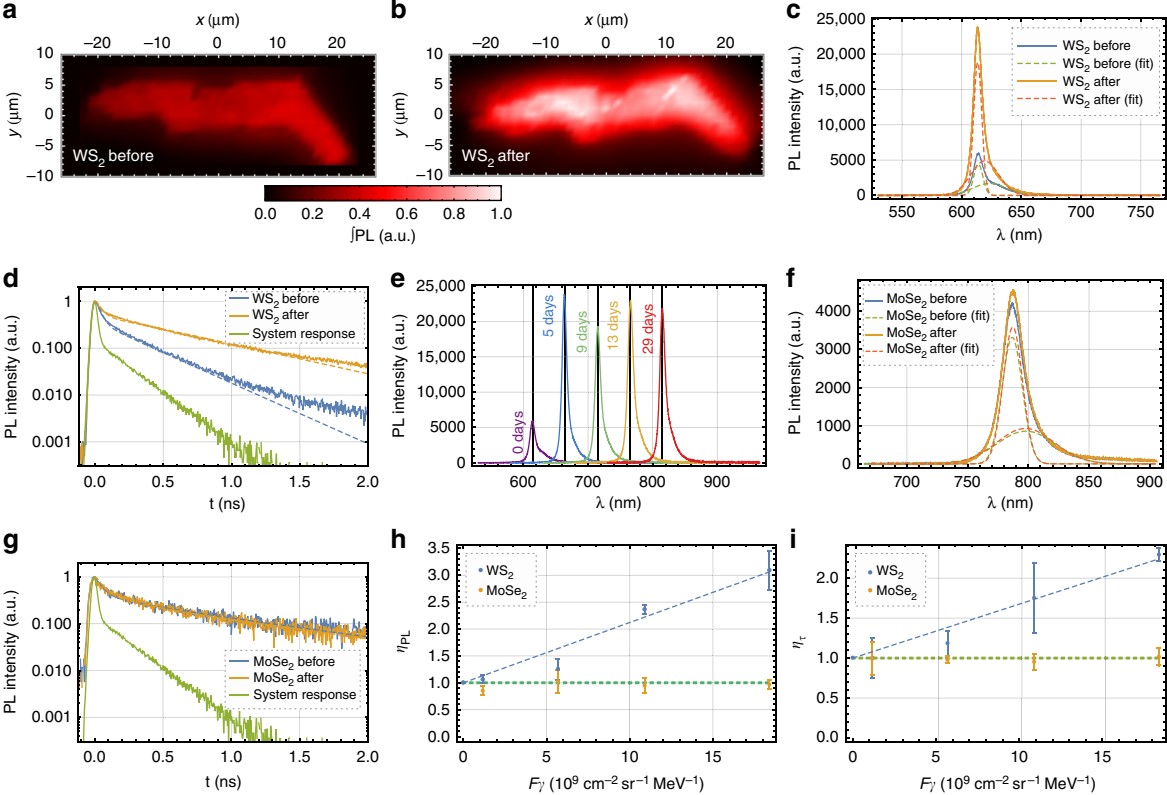

**Fig. 4** γ-Ray tests of transition metal dichalcogenide monolayers. **a–e** Optical characterization of a WS$_2$ monolayer before and after the γ-ray exposure. **a**, **b** The photoluminescence (PL) intensity maps scanned with 1 μm resolution integrated over the full spectrum show a strong increase in brightness after the irradiation. **c** In addition to the brightness increase of 2.99, the PL spectrum shows that the exciton/trion ratio also changed from 0.706(11) to 1.138 (19). This was extracted from fitting two Gaussian distributions. **d** Similarly, the radiative carrier lifetime increased from 336(3) to 678(5) ps. The fit routine deconvolutes the data from the system response. **e** Long-term stability of the PL spectrum measured at different days. The irradiation took place at day 2. For clarity, each subsequent spectra is shifted by 50 nm. The peak wavelength remained invariant with its mean at 614.65 nm. The mean peak wavelength is visualized with the black guidelines. **f**, **g** PL emission spectrum and carrier lifetime for monolayer MoSe$_2$. This material remains predominantly unaffected by the gamma-rays, with the overall brightness increased by <5% and the radiative lifetimes before and after the irradiation being 1086(41) and 1071(47) ps, respectively. Both sample monolayers experienced the same photon fluence. **h**, **i** Relative brightness and carrier lifetime increase as a function of γ-ray fluence averaged over all samples for WS$_2$ and MoSe$_2$. While there is little to no change for MoSe$_2$, for WS$_2$ the relative changes are linearly proportional to the radiation fluence. The data point at zero is the control sample. The green dashed line indicates $\eta = 1$ (no change). The error bars are the standard deviation of the average

enhanced more strongly as the change from $\alpha < 1$ to $\alpha > 1$ shows. This also indicates a change in doping ratio. Given the initial linewidths of 3.80(3) for excitons and 14.07(13) nm for trions, there was no change in center wavelength of the exciton emission (613.89(3) to 613.41(2) nm) and only a slight change of the trion emission (623.45(19) to 619.28(19) nm). However, the linewidths changed to 3.45(2) for excitons and 11.73(13) nm for trions. In addition, the radiative carrier lifetime (see Fig. 4d) had also increased from 336(3) to 678(5) ps (for more data, see Supplementary Note 2 and Supplementary Fig. 4 through 8). The increase in PL and lifetime was persistent over months (see Fig. 4e). Only 1 month is shown, as the samples were subsequently irradiated with protons, but the samples kept their increased PL during these following tests. The small variations in the peak maxima are most likely due to laser defocusing, owing to the small Rayleigh length of the laser with the high numerical aperture objective (see Methods). Nevertheless, quantities independent of this, such as the exciton/trion ratio as well as carrier lifetime, remained fully stable at all measurement days. Moreover, other samples (see Supplementary Fig. 9) were less affected by laser defocusing during the long-term stability tests.

Since free excitons easily scatter and recombine at trapped charge carriers at defect sites, a change in doping ratio as well as

longer carrier lifetime and increased PL intensity likely indicates a reduction in defect density. By averaging over the full data set of samples at the corresponding distance to the source, it can be seen that the effect of an increased PL and lifetime is linearly proportional to the γ-ray flux (see Fig. 4h, i, respectively). Interestingly, this effect was not observed for MoSe$_2$ monolayers, (see Fig. 4f, i). Moreover, under the same exposure conditions, the PL had only increased marginally by 1.05 compared to the 2.99 from the WS$_2$ sample presented previously. In addition, the exciton/trion ratio was stable with $\alpha_i = 1.328(36)$ and $\alpha_f = 1.317$ (36) as well as was the carrier lifetime with $\tau_r^i = 1086(41)$ and $\tau_r^f = 1071(47)$ (see Fig. 4f, g). It should be mentioned that MoSe$_2$ and WS$_2$ have intrinsically different exciton/trion ratios, since our MoSe$_2$ is a p-type and WS$_2$ is an n-type semiconductor. By averaging over all samples, we found $\alpha_{MoSe2} = 1.252(86)$ and $\alpha_{WS2} = 0.715(117)$ (the uncertainty is the standard deviation).

During the data analysis, we noted that the γ-radiation dose was higher than intended due to a calculation error. The resulting highest γ-ray fluence is equivalent to 2170 years at 500 km above the polar caps. However, in terms of space certification this is not an issue. If anything, this further confirms radiation resistance. This proves that 2D materials can withstand even harsher radiation environments than LEO, such as during solar flares or

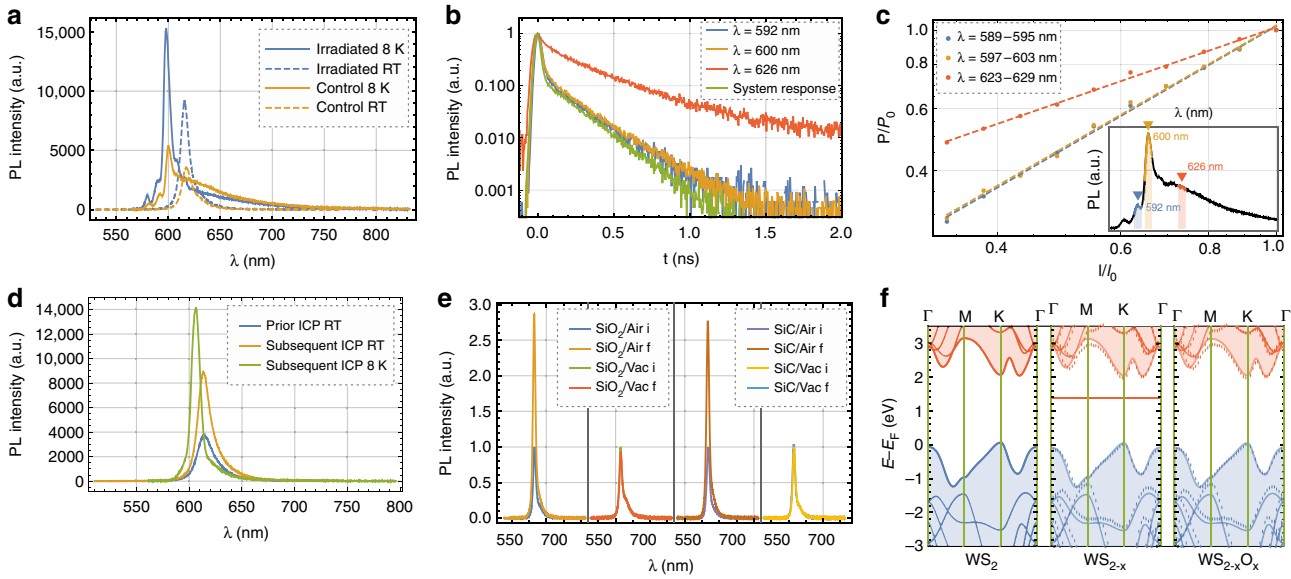

**Fig. 5** Identification of the γ-ray induced healing mechanism. **a** Photoluminescence (PL) spectrum of γ-irradiated and control samples at 8 K and room temperature. The control sample shows strong defect emission in the red sideband. **b** Spectrally and time-resolved PL reveals carrier lifetimes close to the system response at $\lambda = 592$ and 600 nm and 361(3) ps at 626 nm, confirming excitonic and defect nature of the emission. Unlike for defects, the radiative lifetime for excitons/trions is proportional to temperature. The spectral positions at which the lifetimes were measured are marked with correspondingly colored triangles in **c** (small inset). **c** Spectrally resolved power dependence on a log-log plot reveals a slope close to 1 at $\lambda = 592$ and 600 nm, indicating excitonic emission, while the slope <1 at 626 nm means the emission originates from defects. The inset shows the positions in the spectrum (marked with the corresponding colors) at which the power dependence is measured. **d** PL emission prior and subsequent to the inductively coupled plasma treatment shows a similar increase in PL brightness compared to the γ-irradiated samples. In addition, at low temperature no defect emission becomes visible, confirming that oxygen can passivate vacancies. **e** Repetition of the γ-irradiation on $SiO_2$ and SiC substrates as well as in air and vacuum show that the atmosphere must be the source of oxygen used for the defect healing, most likely through adsorbed oxygen onto the surface. **f** Density functional theory calculations of the bandstructure of pristine $WS_2$ (left), $WS_{2-x}$ (middle), and $WS_{2-x}O_x$ (right) show that, unlike the $V_S$ defect, the $S_O$ defect has no unoccupied deep mid-bandgap state. The middle and right bandstructure show the conduction and valence band from the primitive pristine unit cell (solid lines) overlayed with the conduction and valence band from the supercell calculations (dotted lines)

near nuclear reactors. For the $WS_2$ monolayer, we extrapolate the effect of increased PL and carrier lifetime after 4 years in orbit to be less than 0.15% and 0.10%, respectively.

**Backtracing of the healing mechanism.** An increase in PL and carrier lifetime is very surprising: Initially, it was expected that radiation could lead to the formation of new defects, but not to defect healing. We note that low-energy X-ray irradiation of graphene in oxygen environments can lead to the formation of oxygen-related defects[43–45]. Although radiation-induced healing of nanomaterials has been reported[46], such an effect has not been observed with γ-rays, specifically not with such remarkable consequences. It is known that the most common defects in exfoliated TMD materials are chalcogen vacancies[47]. Simulations do also predict that these vacancies can chemically react with oxygen[48], because oxygen itself is a chalcogen. Thus we propose this healing is due to the chemadsorption of atmospheric oxygen, catalyzed by γ-irradiation. A similar mechanism was proposed in a study involving laser-induced defect healing of $WSe_2$[49]. The γ-ray-induced healing observed in our study could happen via several different pathways. One possibility involves the formation of oxygen radicals due to the presence of free electrons from primary reactions like Compton scattering.

To support this, we conducted low-temperature PL measurements of irradiated and control samples at 8 K. The low-temperature environment reduces thermal broadening, which allows the individual emission signature to be resolved. We discovered multiple blue-shifted peaks compared to the RT measurements (see Fig. 5a), most of which are attributed to

negatively charged trions. Consistent with standard semiconductor models and experiments[50], the exciton emission decreases with temperature as the trion emission increases. While both samples exhibit these low-temperature excitonic features, the control sample showed additional PL emission in the red sideband. In contrast, the irradiated sample shows only weak emission in the red sideband. This becomes more evident by comparing the fraction of PL from trions and defects, which is 2.08:1 and 0.35:1 for irradiated and control samples, respectively. Therefore, the defect density had decreased six-fold after the sample was exposed to the γ-radiation. We confirm this by measuring the spectrally and time-resolved photoluminescence (TRPL): The PL emission is coupled to the single-photon counting module via a grating, which makes the TRPL wavelength-selective. Figure 5b shows the lifetime measurements for three wavelengths, with the positions marked with correspondingly colored triangles in the spectrum (see Fig. 5c inset). Unlike for defect states, the radiative lifetime of excitons and trions is directly proportional to the temperature[20]. With the lifetime of the control sample at RT being 286(4) ps, we expect the lifetime of any excitonic emission at 8 K to be around 7 ps. In fact, at $\lambda = 592$ and 600 nm we measured a lifetime just above our system response time (which is ≈3 ps), much shorter than the RT measurements. However, at 626 nm the radiative lifetime was 361(3) ps, thus indicating defect emission. Furthermore, we also measured the spectrally resolved power dependence (see Fig. 5c) at the same wavelengths at which the TRPL was measured. While the slope of the bands around 592 and 600 nm are close to 1 if plotted on a log-log scale, which means it originates from free

excitons or trions, the slope at 626 nm is <1, which indicates defect emission[51].

In the next phase of this study, we confirm that the defect healing is oxygen-related. We replicate the optical signature of the γ-ray-exposed samples by treating freshly prepared monolayers with an $O_2$ inductively coupled plasma (ICP) and optimizing the plasma parameters (see Methods). Figure 5d shows the PL spectrum prior and subsequent to the ICP treatment at RT and 8 K. Much like the irradiated samples, also the monolayers treated with the oxygen plasma show a strong increase in brightness and no defect emission at low temperature as well as a longer carrier lifetime. Similar PL enhancement through defect engineering and oxygen bonding using mild oxygen plasma treatment of monolayer $MoS_2$ has been reported previously[52]. While these results uphold the conjecture of oxygen-related defect healing, the oxygen in the γ-ray experiments could either be supplied by atmospheric oxygen or oxygen from the $SiO_2$ layer. Therefore, we repeated the irradiation with samples on both $Si/SiO_2$ and SiC substrates in air and under vacuum at $10^{-4}$ atm. This will also strongly reduce the amount of surface-adsorbed oxygen. The in-air irradiated samples showed the characteristic increase in brightness and carrier lifetime, while the under-vacuum irradiated samples remained unaffected (see Fig. 5e). We found no dependency on the substrate material. In the context of space certification, this means that $WS_2$ monolayers in evacuated spacecrafts are also not affected even by excessive γ-radiation. In addition to the $WS_2$ crystals, we also exposed $MoS_2$ and $WSe_2$ monolayers to gamma-rays (in air). We observed a slight increase in PL intensity and carrier lifetime after the irradiation for $WSe_2$ (see Supplementary Fig. 8) and no change for $MoS_2$. It should be mentioned that the PL emission from monolayer $MoS_2$ in general is much weaker than for other TMDs, so any change is harder to observe. Furthermore, the change in PL and lifetime for the $WSe_2$ was much weaker than for $WS_2$, even though both samples experienced the same γ-ray fluence.

We now turn to a theoretical analysis of the electronic structure of the proposed defect using DFT. We calculate the electronic bandstructure (see Methods) for pristine $WS_2$, $WS_{2-x}$ (with a sulfur vacancy: $V_S$), and $WS_{2-x}O_x$ (with an oxygen-substituted sulfur atom: $S_O$). The $V_S$ defect has a deep unoccupied state in the bandgap (see Fig. 5f, middle). This is consistent with our experimental observations: The sub-state is an acceptor state trapping electrons, which changes the doping ratio in the crystal. This means more charge carriers are available for charged excitons and thus the trion emission is enhanced. In turn, excitons recombine easily at defects leading to a shorter exciton lifetime. The $S_O$ defect shows no such deep defect state (see Fig. 5f, right), meaning as soon as the vacancy is passivated with oxygen, the electronic configuration is similar to the pristine crystal. The crystal structures of the proposed defects are shown in Supplementary Note 3 and Supplementary Fig. 10.

The $V_{Se}$ and $Se_O$ defects in $MoSe_2$, however, show a very similar electronic structure (see Supplementary Fig. 11), thus our DFT calculations alone cannot explain why the γ-ray-induced defect healing only happens for $WS_2$. Selenide TMDs are known to have less chalcogen vacancies than sulfide TMDs. In fact, in our experiment we see this by the averaged longer carrier lifetime in $MoSe_2$ of 1264 ps, revealing an inherently much smaller presence of defect sites. As already mentioned, scattering and recombination of excitons at defect sites leads to a reduced lifetime. This means intrinsically less defect healing can occur for $MoSe_2$.

**Proton and electron irradiation**. After the γ-ray tests, the samples were irradiated with high-energy charged particles, starting with protons and then electrons. In addition to the γ-irradiated samples, after each radiation test fresh samples were added to study both combined and isolated radiation damage effects. The samples were irradiated with protons from a 1.7 MV tandem accelerator. Owing to the maximally available proton energy of 3.4 MeV, the annual fluence spectra shown in Fig. 1c cannot directly be replicated. Instead, we integrate the annual fluence spectrum over the full energy range for each orbital inclination, which yields 241.820, 721.318, and $464.770 \times 10^6$ cm$^{-2}$ for 20°, 51.6°, and 98°, respectively. Unfortunately, these fluence values are below the range of the used charge carrier counter, which is why we tested the samples at $F_{p^+} = 10^{10}$ cm$^{-2}$. However, at the lower proton energies (200, 500, 1000, and 2500 keV) we used, the potential displacement damage caused by the protons[53] is higher due to the higher stopping power of the 2D materials at lower energies (see Supplementary Note 4 and Supplementary Fig. 12 through 14). As our fluence is anyway higher than required for 500 km altitude, we cannot scale the fluence down according to the used proton energies. For all proton energies, we did not observe any changes in the device performances, PL spectra or carrier lifetimes (see Supplementary Note 5 and Supplementary Fig. 15 through 17). Even after increasing the proton flux 100-fold, there were still no changes. This result is also consistent with previous work[54]. Furthermore, Kim et al. found the onset of degradation at a fluence of $10^{13}$ cm$^{-2}$, with strong degradation at $10^{14}$ cm$^{-2}$[54], while Shi et al. found the damage threshold of $WSe_2$/SiC heterostructures at $10^{16}$ cm$^{-2}$ proton fluence[32]. In our case, the proton fluence of $10^{12}$ cm$^{-2}$ corresponds to 1386 years in orbit (at 51.6° inclination and 500 km altitude). The scaling is based on the number of protons. Hence, we conclude that proton irradiation is no concern for 2D materials and devices in LEO.

Finally, we exposed the samples to electrons using a scanning electron microscope (SEM). The damage to 2D materials caused by electrons is mostly displacement and sputtering[33]. Similar to the proton accelerator, both the energy range as well as the integrated flux are beyond the capabilities of the SEM. The tested energies were 5, 10, 20, and 30 keV while the fluence varied from $10^{10}$ to $10^{15}$ cm$^{-2}$. At 500 km altitude, the integrated fluences are 12.35, 32.06, and $20.73 \times 10^6$ cm$^{-2}$ for equatorial, ISS, and polar orbit, respectively. At the lowest accessible fluence, which is still three orders of magnitude above what is expected in LEO, the crystals were mostly unaffected by the electron irradiation (see Supplementary Note 6 and Supplementary Fig. 18 through 20). Extrapolating the fluence to LEO levels predict that electrons will not have any impact on 2D materials. Higher electron fluences result in permanent loss of PL for TMDs. We propose that this is due to the creation of chalcogen vacancies by knock-on damage, which, as previously mentioned, cause recombination, thereby quenching the PL[55]. However, if the electron energy is increased from 5 to 30 keV, even at the highest fluence such damage was mitigated. This is because higher-energy electrons have a smaller interaction cross-section (see Supplementary Fig. 12). The single-photon emitters remain unaffected by the electron irradiation; however, at extremely high fluences, the emitter density can be increased significantly[17,56]. In our case, this happened while focusing the SEM on a small crystal part before exposing the full crystals to the electrons. The experienced electron fluence at these positions was up to $10^{18}$ cm$^{-2}$ (see Supplementary Fig. 21).

All radiation tests so far indicate that 2D materials tolerate significant amounts of ionizing radiation, far beyond the requirements for LEO. The question arises, if the radiation resilience is also sufficient for higher altitudes. Using SPENVIS, we calculate the integrated annual particle fluence (integrated over the full energy range) as a function of altitude (see Supplementary Fig. 22).

While the proton fluence always remains nearly two orders of magnitude below the damage onset threshold, the electron fluence exceeds the observed damage onset at altitudes >1000 km. It should be mentioned that these calculations assume the same 1.85 mm of Al shielding as above. The shielding also explains the leap in electron fluence at 2000 km: The electron energy increases with altitude and thus actually trapped electrons can penetrate the shielding. As already mentioned, at lower altitudes the electrons are secondary electrons produced via ionizing interactions of high-energy protons with the shielding. Nevertheless, by using an appropriate shield (5.8 mm graded Al/Ta with a Ta-to-Al mass ratio of 35%), the electron fluence can be kept below the damage threshold. Shielding meeting this requirement is common for higher orbits such as in geostationary satellites. Furthermore, this means that 2D materials also can operate in other environments with heavy irradiation, such as during solar flares or near nuclear reactors.

## Discussion

We presented a comprehensive study on the effects of radiation on 2D materials in vision of space certification. Moreover, this study covered the effects of γ, proton, and electron irradiation on TMD-based FETs and single-photon sources in hBN as well as their interaction with blank TMD monolayers. These nanomaterials were investigated back to back, shortly before and after irradiation. While all crystals remained effectively invariant under irradiation relevant for space environments, after excessive γ-radiation monolayer WS$_2$ exhibit significant increase in PL and carrier lifetime proportionally to the photon flux. This is attributed to the healing of sulfur vacancies induced by γ-radiation. We propose that the γ-rays, through a process like Compton scattering, dissociate atmospheric oxygen, which then chemically reacts with the vacancies. This mechanism was confirmed by low-temperature measurements showing that defect emission was weakened upon γ-irradiation. Furthermore, bandstructure modeling of this reaction shows disappearing trapping sites, thus explaining the observed changes.

A potential application of this effect could be a compact radiation dosimeter or radiation detector. In addition to the radiation tests, the low-temperature measurements also confirm that 2D materials survive vacuum and thermal cycling. The tested radiation fluences were much higher than required for LEO. Hence, 2D materials and devices based on them have been proven to withstand the harsh space radiation. Moreover, 2D materials can even operate in environments with heavy irradiation, such as during solar flares or near nuclear reactors. In addition, if the spacecraft shielding is adapted appropriately, we predict that 2D materials can even be used in any orbit. Our results pave the way toward establishing the robustness and reliability of 2D material-based devices for space instrumentation. This combines the fields of space science and nanomaterials, thus opening new possibilities for future space missions.

## Methods

**Device fabrication**. The bulk crystals were acquired from HQGraphene and used as received. After mechanical exfoliation onto Gel-Pak WF-40-X4, monolayer TMD and multilayer hBN crystals were optically identified by contrast and transferred via dry contact to Si/SiO$_2$ substrates (262 nm thermally grown) or 4H-SiC substrates supplied by SiCrystal. The crystal thickness was confirmed using PSI measurements. The hBN crystals were exposed to an oxygen plasma generated from a microwave field at 200 W for 1 min and a pressure of 0.3 mbar at a gas flow rate of 300 cm$^3$ min$^{-1}$ at RT. The subsequent rapid thermal annealing was done under an Argon atmosphere at 850 °C at a gas flow of 500 cm$^3$ min$^{-1}$. The substrates for the FETs have been pre-patterned with gold electrodes using photolithography: After spin coating AZ MiR 701, the positive photoresist is exposed to UV light through a mask and developed. Using electron-beam thermal evaporation, 100 nm of gold is deposited and then LOR 3A was used for lift-off. The monolayer crystals were mechanically transferred between the two electrodes with

an approximate gap of 10 μm, with an attached multilayer crystal touching the electrode completing the electrical connection. The two electrodes served as top gates (source and drain), while the heavily n$^+$-doped silicon substrate served as the back gate.

**Optical characterization**. The home-built μPL set-up utilized second harmonic generation to generate 522 nm ultrashort laser pulses (High Q Laser URDM). The linearly polarized laser is focused down to the diffraction limit by an Olympus ×100/0.9 dry objective. For confocal PL mapping, the samples were moved on Newport precision stages with up to 0.2 μm resolution. The in-reflection collected emission is wavelength filtered (Semrock RazorEdge ultrasteep long-pass edge filter), fully suppressing the pump light, while still collecting the full emission spectrum. This spectrum is recorded using a grating-based spectrometer (Princeton Instruments SpectraPro). The laser pulse length for time-resolved measurements is 300 fs length at a repetition rate of 20.8 MHz. The pulses were split into trigger signal and excitation pulse. The emitted photons were detected by a single-photon counter (Micro Photon Devices) after the grating, so that the time-resolved PL is also spectrally resolved. Both trigger and single-photon signal were correlated by a PicoHarp 300. For low-temperature measurements, a cryogenic chamber was added to the set-up and the samples were cooled down to 8 K with liquid He, at a pressure of 13 μTorr to prevent the formation of ice on the window. The objective was replaced with a Nikon S Plan Fluor ×60/0.7 objective with adjustable correction ring. The second-order correlation function was measured using a Hanbury Brown and Twiss (HBT)-type interferometer in a different confocal set-up with a 512 nm diode laser, equipped with a spectrometer and nano-positioning stage, ensuring that the defects can be localized. The correlation function data are fitted to a three-level system with ground and excited states as well as a meta-stable shelving state:

$$g^{(2)}(\tau) = 1 - Ae^{-|\tau-\mu|/t_1} + Be^{-|\tau-\mu|/t_2} \tag{1}$$

where $t_1$ and $t_2$ are the excited and meta-stable state lifetimes, respectively, $\mu$ accounts for different electrical and optical path lengths in the HBT interferometer, and $A$ and $B$ are the anti-bunching and bunching amplitudes, respectively. The experimental data have been normalized such that for very long time delays, $g^{(2)}(\tau \to \infty) = 1$.

**Electrical characterization**. The FETs were characterized with a Kiethley 4200 Semiconductor Analyzer. One of the the gold electrodes is grounded, while the n$^+$-doped Si substrate functions as a back gate, providing uniform electrostatic doping in the monolayer. Back gate sweeps at different biases between source and drain were measured as well as I–V curves at various back gate voltages. All electrical measurements were carried out at RT.

**Irradiation**. The radioactive isotope $^{22}_{11}$Na was used as a γ-ray source and was kept in a sealed Tungsten container, which was opened for the duration of the exposure. For every disintegration, a 1.275-MeV photon is emitted into 4π. With a branching ratio of approximately 9:1, the decay either happens via a $\beta^+$ transition or electron capture, respectively, resulting in a 90% probability that a positron is emitted. The positrons are shielded by Al foil, where they recombine with electrons to create two γ-rays with energies of 511 keV in opposite directions. The nominal activity was $A = 1.04$ GBq (number of decays), which together with the container geometry leads to a total photon flux of 10.3 MBq cm$^{-2}$ sr$^{-1}$ MeV$^{-1}$ at the output of the container. The differential flux was calculated with

$$F_\gamma = \sum_E \frac{A\eta_E}{G_E E} \tag{2}$$

with branching ratio $\eta_E$, photon energy $E$, and geometrical form factor of the container $G_E$ (which is energy dependent due to the position of the Al foil). The samples were placed at different distances to the source, simulating different altitudes/times in orbit, with a placement accuracy of 1 mm. All samples were mounted facing toward the γ-ray source. The second γ-ray experiment took place 117 days later, after which the source activity decreased to 91.8% ($\tau_{1/2} = 2.603$ years). We accounted for this by adjusting the distance to the source. We exposed samples in air and in a vacuum chamber at 10$^{-4}$ atm. The γ-rays were attenuated by the glass window ports of the vacuum chamber by only 5%. This attenuation does not account for the complete disappearance of the healing effect on the samples in the chamber. For the proton irradiation, a high-energy implanter featuring a 1.7 MV tandem Pelletron accelerator was used. TiH was used as target for the ion sputter source and Ti ions were filtered by a 90° magnet. The tandem accelerator can double the maximal proton energy; however, owing to the used configuration the proton energy was limited from 200 keV to 2.5 MeV. The ion energy is typically well defined within ±5 keV and the error on the fluence is <±10%. The irradiation took place under pressures of 10$^{-7}$ Torr at RT. For the electron irradiation, the SEM from an FEI Helios 600 NanoLab was used, allowing for electron energies ranging from 1 to 30 keV at 2.2 mPa and RT. The current was varied from 0.17 to 0.69 nA. The electron fluence $F_e$ is given by $F_e = \frac{I \cdot t}{e \cdot A}$, where $I$ is the electron current, $t$ the frame time, $e$ the electron charge, and $A$ the frame area. The crystal flakes were located at a very low electron flux and then the SEM was

aligned using another flake nearby, so that the crystal flake under investigation is targeted with a focused electron beam.

**Plasma etching**. We used the commercial ICP-RIE (reactive ion etching) system Samco RIE-400iP and varied all process parameters. We found the optimal process parameters to be 75 W ICP power, 0 W RF power, 3 min plasma interaction time as well as a gas pressure of 6.6 Pa at an oxygen gas flow rate of 30 $cm^3\,min^{-1}$. The RF power is chosen zero to avoid any ion bombardment during the plasma exposure, thus ensuring the process is chemical and not physical (this results in crystal etching or thinning). All ICP processes were carried out at RT.

**Computational methods**. The space environment calculations were performed using the SPENVIS web interface. The proton and electron flux spectra were calculated using the AP-8 MAX and AE-8 MAX models. The shielded fluence spectra for 1.853 mm Al shielding were obtained using the MFLUX package. The interactions between charge carriers and matter are calculated using Monte Carlo simulations (see Supplementary Note 4)[57–59]. These simulations take electromagnetic scattering processes and hadronic nuclear interactions into account. Owing to the more complicated nature of the interactions of electrons with the shielding material, the electron fluence spectra are less accurate. The DFT calculations have been performed using the ab initio total-energy and molecular-dynamics program VASP (Vienna Ab initio Simulation Package) developed at the Fakultät für Physik of the Universität Wien[60,61]. First, the geometry of the pristine conventional cell was optimized using a $15 \times 15 \times 1$ Monkhorst-Pack reciprocal space grid such that all forces were <0.001 eV Å$^{-1}$. We used a plane-wave energy cutoff of 450 eV and norm-conserving pseudopotentials with nonlinear core-correction to describe the core electrons. We also used the Perdew–Burke–Ernzerhof (PBE) functional in the generalized gradient approximation to describe the exchange-correlation energy[62]. The monolayer was constructed using a $7 \times 7 \times 1$ supercell of the optimized primitive unit cell. The ionic positions were then relaxed again, while keeping the cell size fixed. We chose the vacuum distance between each layer, described by the lattice parameter $c$, such that the bandstructure is flat in Γ to A direction of the Brillouin zone. This indicates that there is no inter-layer interaction. We used the same method to obtain the bandstructure of the oxygen and vacancy centers in both, WS$_2$ and MoSe$_2$. These calculations show flat bands in each high symmetry direction, which indicates that there is minimal defect–defect interaction between neighboring supercells. The effective bandstructures shown here were unfolded using the PyVaspwfc package[63,64].

When analyzing these calculations, it is important to remember that PBE DFT systematically underestimates the quasiparticle bandgap[65]. Further, verifying the DFT bandgap against the experimental optical bandgap requires consideration of the exciton binding energy, which is significant in 2D TMDs (~1 eV)[66]. Noting these problems, we only consider our calculations as accurate enough to qualitatively predict the presence and relative ordering of unoccupied defect levels in the bandgap. To confirm our conclusions, future calculations should apply GW corrections.

**Code availability**. The custom code used for analyzing the data of the confocal PL mapping of monolayer 2D materials (averaging algorithm, see also Supplementary Note 1) is freely available and archived at https://doi.org/10.5281/zenodo.2584405.

## Data availability
The data that support the findings of this study are available from the corresponding author upon reasonable request.

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

## Acknowledgements
This work was funded by the Australian Research Council (CE170100012, FL150100019, DE140100805, DE170100169, DE160100098 and DP180103238). We acknowledge financial support from ANU PhD scholarships, the China Scholarship Council and the ANU Major Equipment Committee fund (No. 14MEC34). We thank the ACT Node of the Australian National Fabrication Facility for access to their nanofabrication and microfabrication facilities. This research was undertaken with the assistance of resources and services from the National Computational Infrastructure (NCI). We also thank Hark Hoe Tan for access to the TRPL system. We acknowledge access to NCRIS facilities (ANFF and the Heavy Ion Accelerator Capability) at the Australian National University, in particular we thank Rob Elliman and Tom Ratcliff for assistance with the implanter.

## Author contributions
P.K.L., Y.L. and B.C.B. supervised the project. T.V. and P.K.L. devised the experiments. T.V., K.S., A.S. and L.Z. fabricated and characterized the samples. T.V., P.R. and M.W.D. conducted the theoretical calculations. T.V., J.S. and J.R.M. carried out the irradiation experiments. T.V. and F.K. developed the ICP processes. All authors participated in the result discussions, analysis and the writing of the manuscript.

## Additional information

**Competing interests:** The authors declare no competing interests.

