## [Peer Review File · Nature Communications]

Reviewers' comments:

Reviewer #1 (Remarks to the Author):

The manuscript by Vogl et al. presents a comprehensive study of radiation effects in 2D materials and devices. In my view the valuable contributions of this work reside in (a) its scope, which is relatively broad to include various devices and their components based upon 2D materials, as well as characterization methods (b) study of multiple 2D materials, as opposed to only one material, and (c) the use of various radiation probes. As expected from prior work in the community, the results presented confirm the high tolerance of 2D devices to ionizing radiation. Perhaps the most interesting result reported is the radiation-induced healing of defects, which may actually improve device performance upon radiation. However, please see the comment below which relates to novelty.

The manuscript is well written and suitable for the audience, but there are numerous questions that need to be addressed before it can be considered further.

1. The decrease of ON/OFF ratio is attributed to the temporal variations from surface adsorption. What is the statistical variation of the control samples and irradiated devices, since 80 devices were irradiated? Can this decrease be indeed attributed to radiation by applying standard statistical criteria?
2. Is there a threshold voltage shift of FET devices after gamma or proton irradiation? It seems to be small based upon figures, but would also be suitable to report the exact values or their upper limits.
3. "This proton fluence of 10^{12} cm⁻² corresponds to 1386 years in orbit (at 51.6 deg inclination and 500 km altitude)." When scaling the experimental monoenergetic proton fluence to the space proton environment, is the scaling based on the energy absorbed in the FET (TMD or dielectric), or the number of protons? It is unclear what is meant by "integrate the full spectrum".
4. The proton irradiation result is consistent with the work of Kim, Tae-Young et al., "Irradiation effects of high-energy proton beams on MoS₂ field effect transistors". They did not observe significant change of I-V curves using 10 MeV protons at 10^{12} protons/cm². It would be interesting to compare these results with such previous work. Where would one expect to see the onset of degradation in such an experiment?
5. Photoluminescence enhancement through oxygen adsorption was studied earlier, for example in Nan, Haiyan et al., "Strong photoluminescence enhancement of MoS₂ through defect engineering and oxygen bonding." They showed that mild oxygen plasma can enhance the MoS₂ photoluminescence. This work should be referenced and discussion of such related works is necessary. In light of this, what is the novelty of the reported result?
5. In the section Gamma-ray tests, how was the dose rate determined?
6. "Less is known about these effects on TMDs and other 2D materials" There are recent published works related to ionizing radiation effects on graphene and TMD materials and graphene and TMD FETs, and these should be referenced. See for example the work in Y. P. Chen group at Purdue University on graphene and J. A. Robinson at Pennsylvania State University on TMDs.
7. "It is possible that low energy radiation could change the charge state of the defects in hBN, causing them to enter a dark state. Conversely, high-energy radiation could create new defects in the crystal lattice, which, if close to the quantum emitter could make the emitter unusable." I think the effects are more related to the types of radiation due to the different damage mechanisms, rather than the energy of the radiation. For example, why did the high energy radiation not change the charge state of the defects in hBN? Also, what is considered to be "low" energy here?

8. "From its initial nominal activity of 1.04 GBq, a total photon flux of 10.3 MBq cm⁻² sr⁻¹ MeV⁻¹ emitted into the output mode of the Tungsten container in which the source was kept." It is unclear how the per-unit-energy flux (per MeV) is obtained and whether MBq refers number of decays or number of gamma-rays.

9. Since the results show that the tolerance level is much higher than the radiation environment in LEO, a comment should be made on the radiation resilience for use in Medium Earth orbit, where the radiation level is much higher.

10. In Fig. S1 I appreciate the authors referencing a calculation error in the main manuscript, but what is meant by "the simulated orbit times are 576 times higher than planned"? Which part of this plot does this comment refer to?

11. In Fig. S9 the authors shift the spectrum by 50 nm for each measurement to try to demonstrate invariance of the peak wavelength. This representation is not effective as it is hard to verify that the spectrum did not move. The authors may wish to plot the spectra on multiple plots horizontally or vertically displaced, with guidelines.

12. Fig. S11, "comparizon"

Reviewer #2 (Remarks to the Author):

1. A reasonably large number of papers dealing with radiation effects on 2D materials have been published in the IEEE Transactions on Nuclear Science, including results for graphene, MoS₂, black phosphorus, etc. Some of the results are very relevant for this work, including effects of oxygen on defects. Please compare and contrast these results with those published previously.

2. When testing devices for space applications, it is generally not good practice to irradiate the devices to the specific dose expected in the application environment (or indeed to any one specific dose). The effects may be highly non-linear. It is much better to obtain results at multiple dose (fluence) levels and show that the measured response varies predictably and consistently with the fluence.

3. LEO is a particularly benign radiation environment, which limits the scientific interest of this work. Showing that devices can survive in this environment is a very low bar, indeed. Consequently, this work is of interest primarily as a qualification test of specific devices for this specific environment.

4. Please justify that the tools used for computing radiation dose and damage [47-49] are applicable to 2D materials.

Summary response statement

This statement concerns our revision of the paper entitled *Radiation tolerance of two-dimensional material-based devices for space applications*, based on the reviewer's reports. For the aid of the editor and reviewers, we first provide the reviewer's comments (in blue) followed by a detailed response (in black).

Reviewer #1:

The manuscript by Vogl *et al.* presents a comprehensive study of radiation effects in 2D materials and devices. In my view the valuable contributions of this work reside in (a) its scope, which is relatively broad to include various devices and their components based upon 2D materials, as well as characterization methods (b) study of multiple 2D materials, as opposed to only one material, and (c) the use of various radiation probes. As expected from prior work in the community, the results presented confirm the high tolerance of 2D devices to ionizing radiation. Perhaps the most interesting result reported is the radiation-induced healing of defects, which may actually improve device performance upon radiation. However, please see the comment below which relates to novelty.

The manuscript is well written and suitable for the audience, but there are numerous questions that need to be addressed before it can be considered further.

We would like to thank the reviewer for providing an accurate summary of our work and his/her comprehensive report.

1. The decrease of ON/OFF ratio is attributed to the temporal variations from surface adsorption. What is the statistical variation of the control samples and irradiated devices, since 80 devices were irradiated? Can this decrease be indeed attributed to radiation by applying standard statistical criteria?

We thank the reviewer for requesting this information, this is indeed important to know. The standard deviation of ON/OFF ratio from temporal variations on control samples as well as variations before and after irradiation were roughly 4000. There was negligible difference dependent on the type of radiation for this. We added this information to the manuscript. In addition to the temporal variations caused by surface adsorption, we have seen in our experiments that the I-V curves are highly dependent on the Schottky or contact resistance (the resistance of the contact between

electrodes and the probe tips). This also varies for every contact/measurement. The variations are thus actually caused by both effects. Initially, we only stated this in the supporting information, but also added this to the revised manuscript.

2. Is there a threshold voltage shift of FET devices after gamma or proton irradiation? It seems to be small based upon figures, but would also be suitable to report the exact values or their upper limits.

This is indeed interesting to know. As correctly guessed from the figures, any possible shift of the threshold voltage is small, <0.1 V. We report this upper limit in the revised manuscript.

3. “This proton fluence of 10^{12} cm⁻² corresponds to 1386 years in orbit (at 51.6° inclination and 500 km altitude).” When scaling the experimental monoenergetic proton fluence to the space proton environment, is the scaling based on the energy absorbed in the FET (TMD or dielectric), or the number of protons? It is unclear what is meant by “integrate the full spectrum”.

We thank the reviewer for pointing out this unclarity. Integrating over the full spectrum means taking the annual fluence spectrum shown in Figure 1c and integrate it over the full energy range. The scaling is based on the number of protons: At 51.6° inclination and 500 km altitude the annual proton fluence is 721×10^6 cm⁻², so 10^{12} cm⁻² corresponds to 1386 years in orbit. We clarified this in the revised manuscript.

4. The proton irradiation result is consistent with the work of Kim, Tae-Young *et al.*, “Irradiation effects of high-energy proton beams on MoS₂ field effect transistors”. They did not observe significant change of I-V curves using 10 MeV protons at 10^{12} protons/cm². It would be interesting to compare these results with such previous work. Where would one expect to see the onset of degradation in such an experiment?

We thank the reviewer for this suggestion. Indeed, our proton irradiation tests of the FETs is consistent with the result from Kim *et al.* Moreover, Kim *et al.* show that the degradation starts at fluence values of 10^{13} cm⁻², with significant changes at fluence values of 10^{14} cm⁻². At these fluence values, the defect production sets on significantly, leading to proton-irradiation-induced carrier traps. We added this comparison to the manuscript.

5. Photoluminescence enhancement through oxygen adsorption was studied earlier, for example in Nan, Haiyan *et al.*, “Strong photoluminescence enhancement of MoS₂ through defect engineering and oxygen bonding.” They showed that mild oxygen plasma can enhance the MoS₂ photoluminescence. This work should be referenced and discussion of such related works is necessary. In light of this, what is the novelty of the reported result?

We thank the reviewer for pointing to this work. Oxygen adsorption of 2D materials is indeed known in the 2D community. This can improve the electrical properties (see Ref [49]) or enhance the optical emission, as shown by Nan *et al.* (the work has been cited in the revised manuscript as Ref [52]). The purpose of our plasma experiments was to directly compare the optical emission signature at room temperature and 8 K with the gamma-ray irradiated samples. We show that the irradiated samples have similar emission signatures as the much better understood plasma treated samples. The novelty lies in the fact, that gamma-rays can be used as a catalyst for this oxygen adsorption, which is a very surprising result. Nevertheless, for reasons of practicality, defect healing or improvement of device performance in general should be done via oxygen plasma, rather than using gamma-rays.

5. In the section Gamma-ray tests, how was the dose rate determined?

As this is strongly related to point 8, we answer these questions together below.

6. “Less is known about these effects on TMDs and other 2D materials” There are recent published works related to ionizing radiation effects on graphene and TMD materials and graphene and TMD FETs, and these should be referenced. See for example the work in Y. P. Chen group at Purdue University on graphene and J. A. Robinson at Pennsylvania State University on TMDs.

We agree that especially ionizing radiation effects on graphene have been studied extensively. This includes electron, proton and heavier ion irradiation, as well as X- and gamma-rays. Most of these works look into the modulation of graphene field effect transistors. While there exists less similar work on TMDs and other 2D materials than on graphene, it is true that referencing these articles here is appropriate. We added several references to the revised manuscript. Still, we note that we are unaware of any work looking in combined radiation effects or radiation effects on optical properties of 2D materials.

7. “It is possible that low energy radiation could change the charge state of the defects in hBN, causing them to enter a dark state. Conversely, high-energy radiation could create new defects in the crystal lattice, which, if close to the quantum emitter could make the emitter unusable.” I think the effects are more related to the types of radiation due to the different damage mechanisms, rather than the energy of the radiation. For example, why did the high energy radiation not change the charge state of the defects in hBN? Also, what is considered to be “low” energy here?

The defect can change its charge state for a finite duration, this can occur by absorbing a photon with sufficient energy to promote an electron or hole to/from the conduction/valence band from/to the defect state. Also particle radiation can excite electrons or ionize atoms in the material and the electrons can be captured by the defect. Given the band gap is 6 eV and the defect levels are separated by about 2 eV, the required energy would only be around a few eV (~1-10eV). Photoionization can

occur as long as the photon has sufficient energy to excite an electron/hole to a band, but not too much energy that the photon energy jumps the bands entirely. In this case “low” is considered energies of the order of the band gap. The important thing is that this low energy radiation does not change the material (i.e. no additional defects), which means the defect will eventually return to its original state. For quantum emitters this is known as blinking. Reports of the duration of the “off”-state vary from $< \text{ms}$ to $>10 \text{ s}$. We would not be able to observe this particular effect in our experiments, as this requires *in-situ* monitoring of the emission count rate during the irradiation. The only exception to this is if the promotion of the carrier to the conduction/valence band results in the carrier being emitted by the material or captured by an adsorbate, such that it won't come back.

High-energy radiation can also produce new defects, which can change the charge state permanently. Alternatively, if the new defect is not close enough to change the charge state, but is still within the excitation laser spot, then both defects act as an ensemble of multiple single-photon emitters. We have clarified this in the revised manuscript.

8. “From its initial nominal activity of 1.04 GBq, a total photon flux of $10.3 \text{ MBq cm}^{-2} \text{ sr}^{-1} \text{ MeV}^{-1}$ emitted into the output mode of the Tungsten container in which the source was kept.” It is unclear how the per-unit-energy flux (per MeV) is obtained and whether MBq refers number of decays or number of gamma-rays.

We thank the reviewer for pointing towards this unclarity. The gamma-ray source has a nominal activity of 1.04 GBq. For the later gamma-ray test we adjusted this according to the half-life of ^{22}Na . This number also refers to number of disintegrations, not gamma-rays. 100% of the decays yield a 1.27 MeV photon and with 90% chance a positron is emitted, which recombines into two 511 keV gamma-rays in opposite directions (so maximally one 511 keV photon can hit the sample per decay). That means each decay contributes with $1/1.27\text{MeV}$ and $1/511\text{keV}$ to the per-energy-flux. We assume isotropic emission into 4π . We shielded the positrons and adjusted this the contribution of the 511 keV photons with the position of the shielding. The exact dose rate of $10.3 \text{ MBq cm}^{-2} \text{ sr}^{-1} \text{ MeV}^{-1}$ is then a result of the container geometry. The dose rate decreases with $1/r^2$ with distance r to the source (answer to question 5). We clarified this in the Methods section of the revised manuscript and give an expression for the calculation.

9. Since the results show that the tolerance level is much higher than the radiation environment in LEO, a comment should be made on the radiation resilience for use in Medium Earth orbit, where the radiation level is much higher.

This is actually a really good point. We extended our SPENVIS calculations over both Van Allen belts and found that the proton fluence is always below the damage onset threshold. For electrons, however, the fluence at altitudes $>1000 \text{ km}$ start to exceed the onset damage threshold. These simulations assume the same spacecraft shielding as in the previous simulations. However, at higher altitudes, the satellites will be more

robust and specifically provide thicker and better shielding. Our simulations show, that for a typical 5.8 mm Ta/Al shield (35% Ta to Al mass ratio), also electrons can be sufficiently suppressed. Thus, our study is extended from LEO to all conventional orbits.

10. In Fig. S1 I appreciate the authors referencing a calculation error in the main manuscript, but what is meant by “the simulated orbit times are 576 times higher than planned”? Which part of this plot does this comment refer to?

The calculation error happened during the conversion from hours to days, where the factor of 24 falsely occurred in the numerator rather than the denominator of a fraction (so the difference is a factor of $24^2 = 576$). However, this is not an issue, as this further confirms the radiation tolerance. The plot in Fig. S1 is correct, only without the error the number of years on the top abscissa would be ~ 1 -4 years and the fluence values on the ordinate would be scaled correspondingly.

11. In Fig. S9 the authors shift the spectrum by 50 nm for each measurement to try to demonstrate invariance of the peak wavelength. This representation is not effective as it is hard to verify that the spectrum did not move. The authors may wish to plot the spectra on multiple plots horizontally or vertically displaced, with guidelines.

We agree that in the old plot, verifying that the spectrum did not move was impossible. We believe that the more important information is the peak height though, as the plot shall visualize that the effect on increased photoluminescence was constant over time and did not recover within one month. This can be nicely verified by shifting each spectrum by 50 nm. We also added guidelines of the mean peak wavelength to the plot and explain its use in the figure caption. Indeed, the guidelines indicate negligible changes of the peak wavelength.

We note that this also applies to Fig. 4e (which shows the same long-term stability, but for a different sample).

12. Fig. S11, “comparizon”

We thank the reviewer for the careful reading of our manuscript and pointing out typos. We have edited our manuscript.

Reviewer #2:

1. A reasonably large number of papers dealing with radiation effects on 2D materials have been published in the IEEE Transactions on Nuclear Science, including results for graphene, MoS₂, black phosphorus, etc. Some of the results are very relevant for this work, including effects of oxygen on defects. Please compare and contrast these results with those published previously.

We agree that there exist more papers on radiation effects on 2D materials than we have referenced. This highlights the intense interest in these effects on 2D materials. The vast majority of these papers look into the modulation of field effect transistors with a focus on graphene. Our comprehensive study, in contrast, uses different devices based on different materials. We investigate the influence of multiple radiation types (as well as combined effects) on different material properties, especially also optical properties. Radiation effects on optical properties of 2D materials have, to the best of our knowledge, not yet been studied. Furthermore, we put the study into the context of space qualification by modelling relevant space environments.

In addition, we demonstrate a new gamma-ray induced defect healing effect, which is unheard of so far. We also fully trace-back the mechanism behind this and provide many supporting experiments and theoretical simulations.

Nevertheless, we added multiple references to put our work into context. This also concerns multiple papers in IEEE Trans. Nucl. Sci. which demonstrated that low-energy X-ray irradiation of graphene in oxygen environments leads to the formation of oxygen-related defects and general influence of radiation on FETs.

2. When testing devices for space applications, it is generally not good practice to irradiate the devices to the specific dose expected in the application environment (or indeed to any one specific dose). The effects may be highly non-linear. It is much better to obtain results at multiple dose (fluence) levels and show that the measured response varies predictably and consistently with the fluence.

We thank the reviewer for this insight. Indeed, in order to exclude non-linear effects, longer exposures at lower flux would be suitable. However, we do not believe that non-linear effects play a role for the tested doses. For example ion flux effects should be negligible in our experiments as they rely on defect structures evolving on timeframes longer than the time between ion impacts in the same area. Although the tested proton fluences e.g. are $10^{10} \text{ cm}^{-2} - 10^{12} \text{ cm}^{-2}$, only one in every $10^3 - 10^4$ protons will create a defect. The spacing between the defects is then $> 1 \text{ um}$.

In addition, at the increased radiation flux (compared to space radiation levels) any non-linear effects would increase the possible damage. Since we have not observed damage on the 2D materials, we can confirm that non-linear effects play only a negligible role in our experiments. The only exceptions may be for electrons on TMD materials and gamma-rays on WS₂, where we did see radiation-induced changes. However, similar to the example of the proton irradiation above, calculations show that the probability of having two impacts nearby on timescales of defect creation or lattice excitation is negligibly small.

3. LEO is a particularly benign radiation environment, which limits the scientific interest of this work. Showing that devices can survive in this environment is a very low bar, indeed. Consequently, this work is of interest primarily as a qualification test of specific devices for this specific environment.

We agree with the reviewer that LEO is not a very harsh radiation environment. Nevertheless, the experiments show, that the devices can withstand radiation levels exceeding the requirements for LEO. As requested by reviewer #1, we extended our SPENVIS simulations to MEO and GEO environments and found that with appropriate shielding (as common in these environments), 2D materials will survive as well. Even without additional shielding, proton irradiation would be of no concern. This significantly expands the scope of our work. It is also true, that we tested specific devices, but since we studied different types of devices in different materials we believe that our work is of broad interest and the results apply also to the rest of the 2D material field.

4. Please justify that the tools used for computing radiation dose and damage [47-49] are applicable to 2D materials.

We thank the reviewer for requesting this useful information. In the relevant energy range the radiation damage is created by collisions between the ion and the atoms in the 2D material. SRIM and ESTAR (Ref [47,48] in the old manuscript) should handle this reasonably well. However, the programs use various approximations, including the assumption of a mean free path between scattering events. This approximation should not be valid in the case of monolayered 2D materials. Moreover, appropriate simulations carried out by Lehtinen *et al.* (see *Production of defects in hexagonal boron nitride monolayer under ion irradiation* and *Effects of ion bombardment on a two-dimensional target: Atomistic simulations of graphene irradiation*) showed, that the type of defects in 2D differs from their bulk counterparts. Complex defects are formed due to the recoil of atoms in-plane. The simulations developed by Lehtinen *et al.* are based on the same principles as SRIM and ESTAR.

Nevertheless, the decreasing stopping power with increasing particle energy is at least qualitatively correctly reproduced by SRIM and ESTAR (for the relevant energy range). In addition, the simulations by Lehtinen *et al.* further showed, that defect production probabilities decrease with increasing energies in the MeV range, thus confirming the results from SRIM and ESTAR at least qualitatively. We added this explanation to the Supplementary Information S4, where the simulations are explained.

The calculations of the interactions with the shielding are not affected by this, as they are done for bulk Aluminium. Furthermore, Ref [49] (old manuscript) is only used for calculations involving bulk Al.

REVIEWERS' COMMENTS:

Reviewer #1 (Remarks to the Author):

The authors have adequately addressed the reviewers' comments. I am of the opinion that the manuscript is of sufficient novelty and significance for publication.

Reviewer #2 (Remarks to the Author):

The manuscript is well written and the authors have adequately addressed my comments. Please consider a more general approach in future experiments, based on step-stress testing to identify relevant physical mechanisms.

Summary response statement

This statement concerns our revision of the paper entitled *Radiation tolerance of two-dimensional material-based devices for space applications*, based on the reviewer's reports. For the aid of the editor and reviewers, we first provide the reviewer's comments (in blue) followed by a response (in black).

Reviewer #1:

The authors have adequately addressed the reviewers' comments. I am of the opinion that the manuscript is of sufficient novelty and significance for publication.

We thank the reviewer for his/her time and the assessment of our work.

Reviewer #2:

The manuscript is well written and the authors have adequately addressed my comments. Please consider a more general approach in future experiments, based on step-stress testing to identify relevant physical mechanisms.

We thank the reviewer for his/her time and the assessment of our work. We also appreciate the suggestion of the reviewer allowing us to improve future experiments.